# The Endometrial Microbiota: Challenges and Prospects

**DOI:** 10.3390/medicina59091540

**Published:** 2023-08-25

**Authors:** Pauline Kaluanga Bwanga, Pierre-Luc Tremblay-Lemoine, Marie Timmermans, Stéphanie Ravet, Carine Munaut, Michelle Nisolle, Laurie Henry

**Affiliations:** 1Faculty of Medicine, University of Liège, 4000 Liège, Belgium; kaluangapauline@gmail.com; 2Department of Obstetrics and Gynecology, CHU of Liege—Citadelle Site, University of Liège, 4000 Liège, Belgium; ptremblaylemo@citadelle.be (P.-L.T.-L.); marie.timmermans@chuliege.be (M.T.); michelle.nisolle@citadelle.be (M.N.); 3Center for Reproductive Medicine, University of Liège—Citadelle Site, 4000 Liege, Belgium; stephanie.ravet@citadelle.be; 4Laboratory of Tumor and Development Biology, Giga-Cancer, University of Liège, 4000 Liège, Belgium; c.munaut@uliege.be

**Keywords:** endometrial microbiota, microbiome, fertility, chronic endometritis, uterus, reproductive outcomes, endometriosis, oncology

## Abstract

Contrary to popular belief, we have known for many years that the endometrium is not a sterile environment and is considered to be a low-biomass milieu compared to the vagina. Numerous trials and studies have attempted to establish a valid sampling method and assess its physiological composition, but no consensus has been reached. Many factors, such as ethnicity, age and inflammation, can influence the microbiome. Moreover, it possesses a higher alpha-diversity and, therefore, contains more diverse bacteria than the vagina. For instance, *Lactobacillus* has been shown to be a predominant genus in the vaginal microbiome of healthy women. Consequently, even if a majority of scientists postulate that a predominance of *Lactobacillus* inside the uterus improves reproductive outcomes, vaginal contamination by these bacteria during sampling cannot be ruled out. Certain pathologies, such as chronic endometritis, have been identified as inflammation perpetrators that hinder the embryo implantation process. This pro-inflammatory climate created by dysbiosis of the endometrial microbiota could induce secondary inflammatory mediators via Toll-like receptors, creating an environment conducive to the development of endometriosis and even promoting carcinogenesis. However, studies to this day have focused on small populations. In addition, there is no clearly defined healthy uterine composition yet. At most, only a few taxa have been identified as pathogenic. As sampling and analysis methods become increasingly precise, we can expect the endometrial microbiota to be incorporated into future diagnostic tools and treatments for women’s health.

## 1. Introduction

The human microbiota refers to all the microbial cells present in an individual, whereas the microbiome consists of their genetic inheritance [1]. In general, the microbiota is environment-specific. This is why most studies focus on a single microbiota: the gastrointestinal microbiota, the skin microbiota or others. These microorganisms (bacteria, viruses, fungi, yeast and archaea) are necessary for the physiological state of the organism just as they shape our evolution and phenotype [2]. Indeed, they have a major influence on the immune system. One individual’s ability to prevail against diseases, infections and environmental changes or to maintain homeostasis depends in part on these components [2]. However, it is only in recent years that the scientific community has deconstructed the dogma advocating that the endometrium is a sterile environment. It has been suggested that, like the digestive system, the uterine cavity contains a microbiome necessary to its physiological state [3].

Previously focused on the lower female genital tract, research has shown that *Lactobacilli* species predominate in the healthy vaginal microbiota of women of reproductive age [4,5]. These species produce lactic acid that induces a low-pH environment in the vagina, which impedes the growth of potentially detrimental bacteria and preserves the milieu equilibrium [4]. However, it remains unclear whether microbial populations in the vagina persist in the endometrium. Recent studies aim to describe the endometrial microbiota (EM) in order to compare it to the vaginal microbiota and to draw out any specificities [6]. Sola-Leyva et al. (2021) described the presence in the endometrium of multiple active microorganisms in addition to bacteria, such as fungi, viruses and archaea, at 10%, 5% and 0,3%, respectively [7]. Furthermore, several groups have attempted to establish a correlation between the endometrium environment and physiological or pathological conditions particularly in the field of female infertility [6,8]. To name just a few affections, endometriosis and chronic endometritis (CE) are likely to have an impact on pregnancy outcome. In fact, repeated implantation failure (RIF) and recurrent pregnancy loss (RPL) seem to be more prevalent amongst women suffering from CE [9]. These facts enhance the crucial role of immune and microbial health in a woman’s fertility.

However, some major aspects remain a challenge for the scientific community exploring this field. For instance, there is no consensus yet on the sampling or analysis methods used to study or treat microbial cells of the uterine cavity, leading to diverse results and hypotheses concerning the EM composition and the procedures [5,10]. Indeed, because the lower genital tract is physiologically colonized by microorganisms, all the retrievals passing through the cervix are supposedly at risk of contamination. Furthermore, even though the development of next-generation sequencing (NGS) has enabled more exhaustive analyses of the endometrial microbiota composition utilizing the variable (V) regions of the 16S rRNA gene, the 16S rRNA gene V primers chosen for investigation sometimes vary from one study to another. Therefore, some authors suggest that it might lead to biased results, since depending on the primers, some taxa are representatively misinterpreted [3]. Future objectives would be, firstly, to validate a compliant protocol for the analysis and sampling of endometrial microbiota and, secondly, to establish an accurate treatment to reinstate the uterus eubiosis. These shall be implemented before considering it as a therapeutic option for infertile women.

In this review, we aim to present the current hypotheses on the different aspects of the EM and draw possible future applications.

## 2. Endometrial Microbiota

### 2.1. Composition

Population profiles of the female genital tract (FGT) fluctuate over the course of a woman’s life, depending on age, ethnicity, sexual activity, health status and other factors [4,11,12]. A key feature of the EM is its diversity. Numerous studies have shown that the uterus contains fewer bacteria but more various species than the vagina, resulting in high alpha-diversity (a measure of the diversity of a single sample that is usually based on the number of different species observed) and a low-biomass microbiome [13,14,15].

First, it has been proved that the phyla present in the EM are affected by inflammation [16]. CE, for example, is mostly caused by common microorganisms, such as *Staphylococcus* species, *Escherichia coli (E. coli)* or even *Mycobacterium tuberculosis*, which in some developing countries are responsible for a particular phenotype of CE [9]. Furthermore, the bacteria responsible for acute endometritis are rarely found in chronic endometritis [9].

Second, the microbiome is influenced by the hormonal environment in different ways. The peak of estradiol and the increase in progesterone during the mid-luteal phase are associated with greater microbiota stability in the vagina [11]. Exogenous progesterone, which can be administered, for example, as luteal support after a controlled ovarian stimulation (COS), decreases the diversity of *Lactobacillus* spp. phylotypes and thus modifies the EM [10,17]. Moreover, as the menstrual cycle is characterized by hormonal variations, we can assume that it also shapes the EM where an increased microbial population has been observed during the proliferative phase with a high alpha-diversity that decreases amid the menstrual cycle [7,18,19]. Nevertheless, it appears stable during the few days corresponding to the acquisition of endometrial receptivity [10].

The vaginal microbiome, which does not fluctuate as much as the EM during pregnancy, undergoes changes during delivery [20,21]. Indeed, there is a decrease in the abundance of *Lactobacillus* whereas a stable or higher proportion of this genus in the vaginal microbiota (VM) has been associated with risks of preterm labor [20,21]. McMillan et al. (2015) have even attempted to describe a standard VM composition in pregnant women [22] by comparing their vaginal metabolome to those of non-pregnant women [22]. These examples illustrate the dynamic profile of the FGT microbiota and the difficulty of establishing a benchmark regarding its composition.

### 2.2. Sampling Methods

The recognition of an upper genital tract microbiome is relatively recent, and as previously mentioned, there is no consensus yet about sampling and analysis methodologies. Therefore, the contamination of samples often risks distorting the clinical results of such research. At the beginning, many endometrial samples were collected as fluids with a catheter [23] or a Pipelle [24] introduced through the cervicovaginal canal, and up to 80 µL of endometrial fluid (EF) was aspirated. Some teams first inject 1 mL of collection medium to ensure that the sample correctly reflects the uterine environment [25]. Even if the vagina is cleaned prior to EF collection and the instrument is carefully inserted to avoid contact with the vaginal walls, vaginal or cervical contamination is always possible (Figure 1). This depends mainly on the operator and is, therefore, not always reproducible.

Consequently, recent research has tended to use double-lumen catheters to prevent any contamination. For instance, Reschini et al. (2022) used a double-lumen catheter (Figure 1) with a sampling method requiring three healthcare professionals (a physician, a biologist and a nurse) [5]. Before inserting the outer sheath catheter under ultrasound guidance by a nurse, the physician thoroughly cleaned the cervix and vagina with sterile saline solution. The second internal catheter was then inserted into the first. Next, the biologist performed the aspiration with a 20 mL syringe while the catheter was gradually withdrawn from the cavity, ensuring a more sterile approach [5]. However, Liu et al. (2018) who compared endometrial biopsy (EB) with EF samples suggest that the latter does not fully illustrate the endometrial communities [26]. Indeed, they highlighted the low number of taxa identified per 1000 sequencing reads in EF, the low assortment and regularity of taxa compared to EB samples and the notable differences in predominant species between the two types of samples. However, they recognized that EF bacteria are positively correlated with EB bacteria and suggested that these discrepancies could be related to the attachment or depth of some bacteria to the endometrial walls. All things considered, they recommend using EF as complementary information to the EB study [26]. Moreno et al. (2022) also noted divergences between the constitution of EB and EF [27]. Kitaya et al. (2019) collected EB using a curette, so there was nothing to prevent contamination through the vagina or cervix other than the operator’s abilities [24]. In contrast, Liu et al. (2018) and Moreno et al. (2022) have modified their procedures. They used a Cornier Pipelle, which possesses an outer sheath that reduces the risks of sample contamination from the vagina (Figure 2) [26,27]. The results obtained might also differ according to the DNA extraction kit and the NGS sequencing techniques used. Depending on the hypervariable regions of the bacterial gene encoding the 16S ribosomal subunit chosen to be amplified or depending on the DNA extraction kits, the liability of the results is at stake. All these circumstances play a major role in the difficulty to assess the EM [5,10,15].

## 3. Infertility

Already in 1995, Møller et al. attempted to hypothesize the role of microbes within the uterus [28]. Given that women with distinct EM seem to obtain diverse reproductive outcomes, it is highly expected that the microbial environment of the endometrium influences the pregnancy process [6]. Recently, scientists have become increasingly interested in investigating whether women with RIF, RPL or clinical miscarriages (CM) may have hostile uterine microbiomes mostly colonized by pathogens [27]. Indeed, Moreno et al. suggest that a *Lactobacillus*-dominated (LD) EM increases the rates of successful pregnancies in contrast to a non-*Lactobacillus*-dominated (NLD) EM [6,23]. For example, in this study, patients with a LD microbiota had higher rates of implantation (60.7% vs. 23.1%, *p =* 0.02), pregnancy (70.6% vs. 33.3%, *p* = 0.03), ongoing pregnancy (58.8% vs. 13.3%, *p* = 0.02) and live birth (58.8% vs. 6.7%, *P* = 0.002) [23]. However, as this genus is mainly found in the vagina where it is responsible for its pH and robustness [14,15,29], some studies associate its abundance in the endometrium with contamination of sampling from the vagina or even pathological conditions. Other findings advocate for specific endometrial bacteria, absent in the vagina, hence representing “biomarkers” such as *Stenotrophomonas maltophilia* or *Kocuria dechangensis* [13,30]. Although no bacteria have been systematically associated with uterine dysbiosis, there is evidence that treating uterine pathologies of all types increases the chance of pregnancy [20,31,32]. Seemingly, women presenting any type of pelvic alteration or chronic inflammation, which can also be related to pathogens, strive in order to get pregnant [6]. This highlights the importance of the endometrium health and its microbiome in fertility. In the future, it would be useful to identify and treat women who present symptoms or clinical signs of colonization by pathogens to improve their condition and enhance their fertility.

## 4. Immunology and Chronic Endometritis

Contrary to popular belief in the medical community, the presence of inflammation in the endometrium is not analogous to pathology. Nevertheless, some recent studies suggest that pregnancy should be considered as a global process in which immunological variations are necessary to ensure successful gestation [20,33]. On the one hand, the crucial stages of implantation and placentation require inflammation to allow the trophectoderm to penetrate the endometrial lining [20,34]. Indeed, Gnainsky et al. (2014) demonstrated that an endometrial biopsy enhances its receptivity by attracting inflammatory agents, such as interleukin-6 (IL-6), IL-8, IL-15, CXC-chemokine ligand 1 or osteopontin and tumor necrosis factor (TNF) [35]. On the other hand, implantation, placentation and subsequent stages also necessitate immunomodulation and tolerance with great involvement of T-regulatory cells. In particular, a lack of tolerance can lead to preeclampsia due to insufficient blood supply to the fetus [36]. During fetal growth, the environment evolves towards a T-helper type 2 (Th2)/anti-inflammatory milieu for the longest period of pregnancy with an increased population of macrophages and natural killer (NK) decidual cells. Finally, to activate labor, a pro-inflammatory environment is once again necessary [20].

However, certain pathologies, such as chronic endometritis, have been identified as inflammatory factors that interfere in this process. First, chronic endometritis is defined as a prolonged state of inflammation of the endometrium characterized by the presence of edema, increased stromal cell density and dissociated maturation of the stroma and epithelium throughout the menstrual cycle [37]. These alterations are capable of causing dyspareunia, chronic pelvic pain or even abnormal uterine bleeding, to name a few [38], and are generally correlated with plasma cell infiltration in the endometrial stroma area (ESPC). Here again, neither the diagnostic criteria nor the methods are clearly established, but as mentioned above, most studies associate this pathology with ESPC, which is detected with immunohistochemistry (IHC) staining. This technique searches for the transmembrane heparan sulfate proteoglycan syndecan 1 (CD138), which is a well-known marker of the ESPC. Bouet et al. (2016) conducted a study in which they combined office hysteroscopy with IHC to improve the diagnosis of CE and found that office hysteroscopy is an interesting tool but cannot be used exclusively for this indication [38]. Like other studies, they identified a higher prevalence of CE in women suffering from RPL and RIF or encountering obstacles to achieve a successful pregnancy, demonstrating that excessive inflammation leads to infertility [9,38,39]. There is no single bacterium attributed to the genesis of CE; those detected may be common or pathological bacteria that, therefore, cause dysbiosis of the EM and lesions. For example, *E. coli, Mycoplasma/Ureaplasma* species, *Pseudomonas aeruginosa*, *Gardnerella vaginalis* or *Corynebacterium* have already been involved [20]. The reported microbiota cannot be assimilated to non-CE EM or to any other suggested healthy composition (LD or NLD) [20] and, above all, show significant signs of inflammation. Although there is no international recommendation for treatment yet, oral antibiotic therapy is commonly used and has been shown to be effective in eliminating ESPC [20]. It is, therefore, essential to take into account the immunological and inflammatory aspects of the endometrium and its microbiota during infertility assessments.

## 5. Endometriosis

In recent years, there has been growing evidence that microbial colonization of the endometrium has an impact on the pathogenesis of endometriosis. Indeed, a pro-inflammatory climate created by bacterial endotoxin or lipopolysaccharide is thought to induce secondary inflammatory mediators (like nuclear factor-κB (NF-κB)) via Toll-like receptors in the peritoneal cavity, creating an environment conducive to the development of endometriosis [40,41]. The abundance of Gram-negative bacteria in the microbiota of patients with endometriosis would support this theory [40]. In addition, *E. coli* has been found more frequently in the menstrual blood and endometrial smears of women with endometriosis than in the control, reinforcing the hypothesis of bacterial contamination [42,43]. A study by Tai et al. in 2018 showed an increased risk (HR 3.02) of endometriosis in patients with pelvic inflammatory disease (PID), adding to the idea of an intricate relationship between inflammation, dysbiosis and endometriosis [44].

Moreover, bacteria inside the uterus are more diverse in people with endometriosis [45,46]. They contain fewer *Lactobacillus* species than controls [40]. Whether this diversity is more prone to a pathological state is not yet known, but it could be a hallmark of endometriosis and a diagnostic tool. A recent study found that there are significant differences in the cervical microbiota of patients with endometriosis [45]. These differences could be a diagnostic indicator of endometriosis. Furthermore, a recent study by Perrotta et al. found that certain profiles of the vaginal microbiome could be specifically linked to certain stages of endometriosis [47].

This relationship between microbiota and endometriosis could also lead to specific treatments for this disease. In a recent study by Chadchan et al., endometriotic lesions were significantly smaller in mice treated with broad-spectrum antibiotics than controls with fewer proliferating cells [48]. Similarly, probiotics, which are live organisms that can positively modify the microbiota when ingested, might be a way of treating the disease [49].

## 6. Oncology

Oncology is another medical field in which the endometrial microbiota could play an important role. Infectious diseases and their involvement in the development of certain types of cancer are well known through various pathways: genetic mechanisms, chronic inflammation and epithelial injury. *H. pylori*, for instance, promotes gastric cancer by inducing chronic inflammation, thus creating bacterial proliferation and, subsequently, the conversion of nitrates by bacteria into carcinogens [50]. Although mice infected only with *H. Pylori* do not develop more tumors or even fewer than their pathogen-free counterparts, this bacterium acts as a promoting agent of a more complex microbiota, thus leading to the previously mentioned carcinogenic effect [50].

The microbiota via its altered state, called dysbiosis, is suspected to promote carcinogenesis by altering the host immune defense responses [51], shifting the balance between cell proliferation and death and influencing the metabolism of self-produced factors, ingested molecules and drugs [52,53]. As an example, NF-κB, a key regulator of inflammation in cancer, has been shown to be activated by certain bacteria (such as *F. nucleatum* in colorectal cancer) through the activation of Toll-like receptors and nucleotide-binding oligomerization domain-like receptors [54].

Walther-António et al. described that a high vaginal pH, a hallmark of dysbiosis, is associated with endometrial cancer (EC) [55]. In their study, specific bacteria, such as *Firmicutes*, *Spirochaetes*, *Actinobacteria* and *Proteobacteria*, were identified in women with EC, suggesting that a different microbiota may be associated with a carcinologic condition of the uterus. In a study by Lu et al., *Micrococcus* was associated with endometrial microbiota dysbiosis and inflammatory cytokines (such as TNF-α, IL-6 or IL-8) in patients with EC [56]. These findings could be helpful for future research that explores the relationship between endometrial microbiota, inflammatory responses and cancer [56].

The efficacy of cancer treatment is also impacted by the microbiome, as shown by various types of malignancies, such as melanoma [57].

## 7. Future Prospects

Clearly, the female genital tract microbiota has emerged as a field of growing interest in reproductive and immune medicine. Today’s knowledge holds out the promise of significant future clinical implications, not least in the precise recognition of associated endometrial or vaginal dysbiosis but also in their appropriate treatment. This might also allow for more personalized medicine that can bring better care to patients. Certainly, it could be, for instance, first implemented in assisted reproductive technology, where VM and EM studies shall be instated as available initial procedures when first establishing the patient profile. Furthermore, this could be investigated in gynecology for women suffering from chronic endometritis or even endometriosis, as it would allow for the determination of the pathogens in cause and thus apply a selective treatment. Moreover, as stated above, a better microbial comprehension and regulation could be a tool used in oncology. As dysbiosis disturbs the organism’s immune response, and some bacteria may promote carcinologic conditions [55,56], searching for and treating an EM could improve the efficacy of cancer treatment [57].

## 8. Conclusions

Research into the importance of the endometrial microbiota in gynecology is increasing exponentially. However, many aspects still need to be elucidated before a consensus can be reached. The characterization of a normal microbiota is not yet established, as is the case for *Lactobacilli*. Only certain types of bacteria have been identified as being associated with a pathological or healthy state. The sampling method also needs to be standardized and improved to avoid potential contamination. New technical approaches have been developed to overcome this obstacle. Moreover, analysis of the microbiota using NGS rather than culture-based methods should be the norm in order to detect all taxa within a sample. Finally, one of the limitations of recent studies concerns the size of population samples, which are still relatively small.

Prospective studies on larger populations with strict inclusion and exclusion criteria are needed to investigate the different fields of applications with a strong clinical perspective. A thorough understanding of the interactions between the microbiota and the host will potentially open up new avenues for prevention, diagnosis and future therapies.

## Figures and Tables

**Figure 1 medicina-59-01540-f001:**
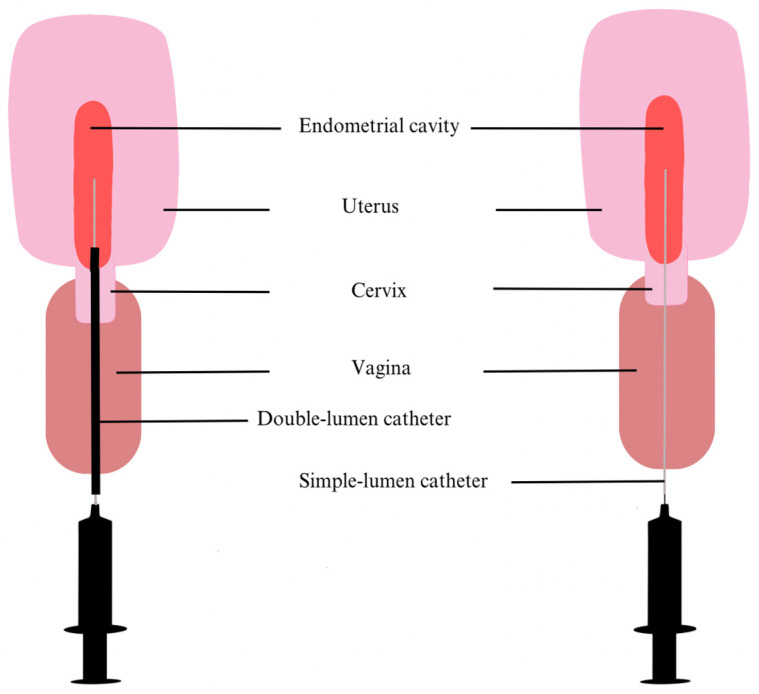
The two EF sampling methods. The double-lumen catheter on the left has a hollow outer sheath, shown here in black, which covers the smaller gray inner catheter, reducing the risk of contamination from the vagina. The external part is inserted through the cervix with its extremity just beyond the internal ostium through which the smaller catheter reaches the endometrial cavity. The extra protection is missing in the right figure, where the single-lumen catheter passes directly through the cervix into the endometrial cavity to collect the EF.

**Figure 2 medicina-59-01540-f002:**
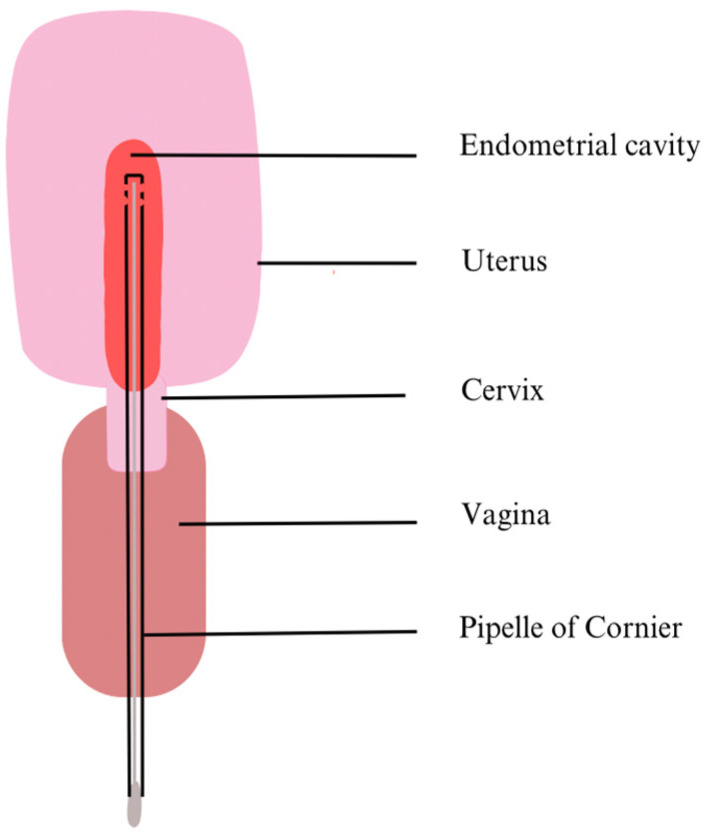
EB sampling method. The Cornier Pipelle is a curette into which an aspiration device is inserted. When it reaches the uterine cavity, the sample of endometrial mucosa is collected by aspiration.

## Data Availability

Not applicable.

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
