# Peer review of "The Endometrial Microbiota: Challenges and Prospects"

_medicina, 2023, doi:10.3390/medicina59091540_

Round 1

Reviewer 1 Report

Work by Kaluang Bwang et al. fri "The endometrial microbiota: challenges and prospects" is an interesting literature item that requires a few changes for better reception:

- please explain all abbreviations used in the text (I understand that EM stands for endometriosis, but please ensure that all abbreviations used are fully developed in the paper);

- please mention in the text in the introduction chapter about the problems and challenges faced by specialists and scientists in the issue raised by the authors, which will allow you to outline the next chapters in the text (after all, we have challenges and prospects in the title)

- please move Figure 1 closer to the place of its citation, it will make it easier for the reader to refer to the presented figure when reading this paragraph of the text;

- the same remark as above applies to Figure 2, this will additionally avoid a large amount of space on the page, without reducing the size of the figure;

- please also consider changing the organization of the text, because in the current version, it is a bit unclear. I suggest that after the introduction, the chapter Composition of the endometrial microbiota together with Sampling methods be merged, because in my opinion these two aspects are extremely interrelated, then introduce the chapter on the role of microbiota in endometriosis; then the role of the microbiota in Chronic Endometritis; next, it is worth discussing the role of microbiota in infertility a bit more broadly; then discuss the interactions of the microbiota with the immune system, and finally discuss its importance in oncology.

-I also suggest adding a chapter on future perspective, because it is missing a lot, and as the title says, it is quite important.

-in conclusion, I propose to emphasize the challenges faced by the endometrial microbiota examination

Author Response

First of all, I would like to thank you for taking the time to read and respectfully review our article in order to help us improve it.

1) All the abbreviations have been verified and EM is standing for Endométrial Microbiota

 2) A few lines have been added to explain the challenges faced by specialists and scientists (various sampling methods, different primers studied, no consensus..)

3&4) Figures 1&2 have been moved

5) We merged paragraphes 2 and 3 under 2. Endometrial Microbiota/ 2.1 : Composition / 2.2 Sampling methods and discussed a bit more broadly the impact of EM in infertility where, at the same time, we introduces the inflammatory diseases just as Chronic Endometritis and endometriosis 

6) A paragraph entitled “Future Prospects” has been added.

7) We globally emphasised the challenges faced by the scientists by explaining them more into details and also by suggesting the requirements for future clinical use.

Reviewer 2 Report

I read an interesting paper on endometrial microbiota. The topic is very interesting and certainly promiscuous. However, the authors did not shy away from some shortcomings, which I will list in points:

       1.- “They used a Cornier Pipelle which possesses an outer sheath that preserves better sample sterility” it is hard to talk about “sterility” if you are sampling bacteria. The term is misused.

        2.  - Figure 2 have to be corrected

         3. -  “In the future, it would be useful to identify and treat women undergoing IVF treatments who present symptoms or clinical signs of colonization by pathogens.” I do not understand why only women in IVF programs should be treated. If you diagnose and treat infertile women well, they in many cases do not need IVF for reproductive success.

         4. -  Sections are missing:

-          “Author Contributions:

-          Funding:

-          Institutional Review Board Statement:

-          Informed Consent Statement:

-          Data Availability Statement:"

NA

Author Response

First of all, I would like to thank you doctors and specialists for taking the time to read and respectfully review our article in order to help us improve it.

1) the sentence has been modified into “that reduces the risks of sample contamination from the vagina”

2) Figure 2 has beencorrected*

3) Indeed, the sentence has been changed to state that it will be offered to every women.

4) Sections have been filled.

Reviewer 3 Report

The article "The endometrial microbiota: challenges and prospects" by Pauline Kaluanga Bwanga et al. provides a systematic review of current knowledge on the endometrial microbiota.

The article is well organised, clear and concise. The English used is correct. The abstract adequately summarises the content of the article. Finally, the bibliography is quite extensive (as befits a literature review) and complete.

Although it is a field in which we still do not know much, I think the authors have done a very good job in summarising the current knowledge and establishing a good starting point for future work.

My congratulations to the authors

Author Response

Thank you doctor/specialist for taking the time to read our paper and giving us a respectful and honest review. Thank you for your appreciation.

Round 2

Reviewer 1 Report

Thank you very much for responding to my comments